

**Conceptual models of dissolved carbon fluxes considering**
**interannual typhoon responses under extreme climates in a**
**two-layer stratified lake**
Hao-Chi Lin[1], Keisuke Nakayama[2], Jeng-Wei Tsai[3], and Chih-Yu Chiu[4]
[1] Department of Geography, National Taiwan University, Taipei City, Taiwan
[2] Graduate School of Engineering, Kobe University, Kobe City, Japan.
[3] Department of Biological Science and Technology, China Medical University, Taichung
City, Taiwan.
[4] Biodiversity Research Center, Academia Sinica, Taipei City, Taiwan.
* Corresponding author: Keisuke Nakayama (nakayama@phoenix.kobe-u.ac.jp)



**Abstract**

15       Extreme climates affect the seasonal and interannual patterns of carbon (C) distribution

due to the regimes of river inflow and thermal stratification within lentic ecosystems. Typhoons
rapidly load substantial amounts of terrestrial C into subtropical small lakes, renewing and mixing
the water column. We developed conceptual dissolved C models and hypothesized that
allochthonous C loading and river inflow intrusion may affect the dissolved inorganic C (DIC)
and dissolved organic C (DOC) distributions in a small subtropical lake under these extreme
climates. A two-layer conceptual C models was developed to explore how the DIC and DOC
fluxes respond to typhoon disturbances on seasonal and interannual time scales in a small
subtropical lake (i.e., Yuan–Yang Lake) while simultaneously considering autochthonous
processes such as algal photosynthesis, remineralization, and vertical transportation. Monthly
field samplings were conducted to measure DIC, DOC, and chlorophyll $a$ concentrations to
compare the temporal patterns of fluxes between typhoon years (2015–2016) and non-typhoon
years (2017–2018). The results demonstrated that net ecosystem production was 3.14 times higher
in the typhoon years than in the non-typhoon years in Yuan–Yang Lake. The results suggested
that the load of allochthonous C was the most crucial factor affecting the temporal variation of C
fluxes in the typhoon years; on the other hand, the transportation rate shaped the seasonal C in the
non-typhoon years due to thermal stratification within this small subtropical lake.



## 1.  Introduction

The Intergovernmental Panel for Environmental Changes Sixth Assessment Report
(IPCC AR6) (2021) suggested that, by 2050, not only is air temperature going to increase by at
least about 1.5 °C but high-intensity storms and drought events will become more frequent as a
result of global warming and climate change. In freshwater ecosystems, extreme climates may
change the mixing regimes of water columns (Kraemer et al., 2021; Maberly et al., 2020;
Woolway et al., 2020), heat wave events (Woolway et al., 2021a; Woolway et al., 2021b),
droughts (Marcé et al., 2019), and floods (Woolway et al., 2018). Freshwater ecosystems store
around 0.32 to 1.8 Pg C yr$^{-1}$, which is approximately equivalent to shallow coastal areas; these
ecosystems provide important services for human sustainability, such as acting as processing
hotspots in regional C cycling (Aufdenkampe et al., 2011; Cole et al., 2007; Engel et al., 2018;
Lauerwald et al., 2015; Raymond et al., 2013). Extreme weather events might induce stronger
seasonal thermal stratification from spring to summer and longer overturn from autumn to
winter, thereby changing the C distribution and transportation within water bodies (Kraemer et
al., 2021; Olsson et al., 2022a; Woolway et al., 2020). The responses of C fluxes in small lakes
(lake area < 1 km$^2$) are sensitive to climate change due to the ease with which these C mix with
water columns (Doubek et al., 2021; MacIntyre et al., 2021; Winslow et al., 2015). Moreover,
storms induce dramatic changes in thermal stratification and water inflows (Lin et al., 2022;
Olsson et al., 2022b; Vachon and Del Giorgio, 2014; Woolway et al., 2018). River inflows and
wind turbulence mix the allochthonous C from sediments into the water column after storm
events in small stratified lakes (Bartosiewicz et al., 2015; Czikowsky et al., 2018; Vachon and
Del Giorgio, 2014). However, small lakes account for 25% to 35% of the total area of the earth's
surface lakes (Cole et al., 2007; Downing et al., 2006; Raymond et al., 2013). However,
compared to the case in larger lakes, C fluxes in small lakes remain uncertain because small
lakes have usually been ignored in calculations of C flux on a global scale (Cole et al., 2007;
Raymond et al., 2013). Thus, elucidation of the C fluxes in small lakes in extreme climates
would be key to optimizing the estimations of global C fluxes in extreme climates.
Understanding the influences of physical, hydrological, and biogeochemical processes
on the fates of C fluxes in smaller lake ecosystems is challenging work (Aufdenkampe et al.,
2011; Cole et al., 2007; Raymond et al., 2013; Tranvik et al., 2009; Vachon et al., 2021;
Woolway et al., 2018). This is not only because of difficulties in measurement but also because
of the dynamics and interactions between factors and processes associated with C fluxes.
Dissolved inorganic carbon (DIC) concentration is an important factor in estimating $CO_2$ fluxes
within lake ecosystems (Smith, 1985). Among C fluxes in a freshwater body, the practical
pressure of $CO_2$ ($pCO_2$), defined as $CO_2$ emission across the air–water interface, is affected by
DIC, water temperature, wind speed, and pH (Jähne et al., 1987; Smith, 1985). River inflows,



sediment, and respiration contribute to DIC loading into lakes (Hope et al., 2004; Vachon et al.,
2021); simultaneously, autotrophic organisms, such as planktons and submerged vegetation,
capture DIC via photosynthesis (Amaral et al., 2022; Nakayama et al., 2020; Nakayama et al.,
2022). Moreover, calcification and mineralization may consume dissolved oxygen within water,
inducing uncertainty in $p$CO$_2$ estimation (Hanson et al., 2015; Lin et al., 2022; Nakayama et al.,
2022). Dissolved organic carbon (DOC) might contribute to CO$_2$ emission from lake water to
the atmosphere through mineralization and remineralization within lake ecosystems (Hanson et
al., 2015; Sobek et al., 2005). In subtropical freshwater ecosystems, DOC concentration is one
of the vital factors in describing variances in mineralization and remineralization rates for
dissolved C (Lin et al., 2022; Shih et al., 2019). Kossin et al. (2013) investigated global storm
events with an accumulated rainfall of about 50 mm, which is approximately 10% to 40% of
precipitation in a subtropical typhoon event. Other studies found that typhoon disturbances
quickly mix, renew, or dilute the water in small subtropical lakes (Ejarque et al., 2021; Kimura
et al., 2012; Kimura et al., 2017; Lin et al., 2022). Therefore, investigation of the magnitudes of
DIC and DOC during typhoon disturbances is essential to understand the seasonal regimes and
to estimate C fluxes in small subtropical lakes.
Typhoons' effects on C fluxes were studied in a small, two-layer stratified, subtropical
lake, Yuan–Yang Lake (YYL) (Chiu et al., 2020; Jones et al., 2009; Lin et al., 2021; Lin et al.,
2022). Jones et al. (2009) used the conceptual hydrology model and sensor data to estimate CO$_2$
emission in YYL during the typhoon disturbances that occurred in October 2004: 2.2 to 2.7 g C
m$^{-2}$ d$^{-1}$ of CO$_2$ was released into the atmosphere. CO$_2$ emissions into the atmosphere were
recorded at around 3.0 to 3.7 g C m$^{-2}$ d$^{-1}$ because of substantial loads of terrestrial C via river
inflows after strong typhoons in YYL (Chiu et al., 2020). Especially, vertical mixing, thermal
stratification, and river retention regimes were essential physical processes in the C fluxes in
YYL (Lin et al., 2021; Lin et al., 2022). These studies suggested that river intrusion and thermal
stratification are key factors shaping the seasonal and interannual patterns of C fluxes during
typhoon disturbances. River intrusion not only controlled the C fluxes, algal biomass, and
nutrient loading, but also influenced the length of stratification and hydraulic retention times
(Lin et al., 2021; Lin et al., 2022; Maranger et al., 2018; Nakayama et al., 2020; Olsson et al.,
2022a; Olsson et al., 2022b; Zwart et al., 2017; Vachon and Del Giorgio, 2014). We
hypothesized that allochthonous C loading and river inflow intrusion may affect the DIC and
DOC distributions (**Figure 1**). At the same time, autochthonous processes in small subtropical
lakes, such as algal photosynthesis, remineralization, and vertical transportation, must also be
considered (**Figure 1**). Here, we followed our hypothesis to develop two-layer conceptual C
models to assess C flux responses to typhoon disturbances in small subtropical lakes.



## 2. Materials and methods

### 2.1 Study site

YYL is a shallow (mean water depth: 4.3 m) and oligotrophic (total phosphorous: 10-20 µg-P L$^{-1}$; total nitrogen: 20-60 µg-N L$^{-1}$) subtropical mountain lake (Chou et al., 2000; Tsai et al., 2008; Wu et al., 2001) on Chi-Lan Mountain at around 1,640 asl in north-central Taiwan (24.58° N, 121.40° E) (**Figure 2**). Its water is brown because of its humic acid content (colored dissolved organic matter: 20-50 ppb QSE; mean pH: 5.4). YYL is surrounded by old-growth trees such as *Chamaecyparis formosensis*, *Chamaecyparis obtusa* var. *formosana*, and *Rhododendron formosanum* Heiml (Chou et al., 2000). The annual precipitation is over 3,000 mm yr$^{-1}$, and typhoon precipitation contributes up to half of the total precipitation in YYL (Chang et al., 2007; Lai et al., 2006). Due to rapid renewal of the water body, the water retention time (or residence time) was around 4.4 days in typhoon Megi from 27 September to 1 October 2016 (Lin et al., 2022). The water surface temperature ranges from 15 to 25 °C during March to August, and the water column overturns in September (Kimura et al., 2012; Kimura et al., 2017; Lin et al., 2021). The concentrations of dissolved C (Lin et al., 2021), nutrients (Chiu et al., 2020; Tsai et al., 2008) and organisms (Shade et al., 2011) increase within YYL from autumn to winter. YYL is registered as a long-term ecological study site by the Ministry of Science and Technology (MOST) of Taiwan since 1992 and became part of the Global Lake Ecological Observatory Network (GLEON) in 2004.

### 2.2 Water sampling and chemical analysis

We collected water quality samples (DOC, DIC, Chl. *a*) at water depths of 0.04, 0.50, 1.00, 2.00, and 3.50 m at a buoy site (**Figure 2**). We also measured the water surfaces of six river inflows and an outflow monthly using a horizontal van Dorn bottle (2.20 L, acrylic) from January 2015 to December 2018 (**Figure 2**). These samples were collected by using a portable hand pump and glass microfiber filter papers (47 mm GF/F, nominal pore size 0.70 µm; Whatman, Maidstone, Kent, UK) to obtain filtrate samples. Water samples were stored at around 4°C in a refrigerator until analysis. Samples were analyzed by using an infrared gas detector to detect DIC and DOC concentrations with persulfate digestion (model 1088 Rotary TOC autosampler; OI Analytical, College Station, TX, USA). The filter papers were kept in opaque bottles at around −25 °C in a refrigerator until the samples were analyzed by using a portable fluorometer (model 10-AU-005-CE; Turner Designs, Sunnyvale, CA, USA). In the laboratory, the filter papers were extracted with methanol to obtain the Chl. *a* concentration. These samples were analyzed for less than 72 h to prevent light and chemical degradation.

### 2.3 Data analysis and numerical modeling



Three water quality variables (DIC, DOC, and Chl. *a*) were compared between different
layers (upper and lower layers), years (typhoon years and non-typhoon years), and seasons (spring,
summer, autumn, and winter). First, we separated our investigation data into typhoon years and
non-typhoon years as described in Sect. 2.3.1. Next, we developed a conceptual equations model
to generate continuous DIC and DOC data at the upper and lower layers as shown in **Figure 1**.
This helped us understand the transportation, photosynthesis, and remineralization rates between
seasons and between typhoon and non-typhoon years (Sect. 2.3.2).

*2.3.1    Typhoon and non-typhoon years*
We collected meteorological data from a meteorological tower located about 1.0 km
from YYL (Lin et al., 2021; Lin et al., 2022). Rainfall (model N-68; Nippon Electric Instrument,
Tokyo, Japan) and wind speed (model 03001, R.M. Young, Traverse City, MI, USA) data were
stored in a datalogger (model CR1000; Campbell Scientific, Logan, UT, USA) for every 10 min.
River discharge ($Q_{in}$, m$^3$ d$^{-1}$) was estimated using the rainfall data and a water depth meter (model
HOBO U20; Onset Computer, Bourne, MA, USA) at the end of a river inflow (**Figure 2**) for
every 10 min by following the Manning formula. Transparency was estimated by using Secchi
disc data measured at local times (GMT+08:00) from 10:00 to 14:00.
As **Table 1** shows, four strong typhoons were recorded, contributing a total of 2,254
mm of precipitation in all 24 months of 2015 and 2016, accounting for 35.6% of annual
precipitation. However, there no typhoon rainfall was recorded at YYL in 2017 and 2018; the total
precipitation in that 2-year period was around 2,537 mm. There was no significant difference in
average water depth between 2017 and 2018 (**Table 1**). The averaged discharge was less than 774
m$^3$ d$^{-1}$ in 2017 and 2018. Thus, we considered 2015 and 2016 as typhoon years, and 2017 and
2018 as non-typhoon years.

*2.3.2    Conceptual two-layer DIC and DOC model*
Nakayama et al. (2010) successfully developed a conceptual two-layer dissolved oxygen
model based on the strong wind turbulence at Tokyo Bay. Additionally, Lin et al. (2021) pointed
out that thermal stratification that inhibits vertical C flux between the upper and lower layers in
shallow stratified lakes make it possible to develop conceptual two-layer C models (Lin et al.,
2022; Nakayama et al., 2022), and the phytoplankton and remineralization effects on DIC and
DOC fluxes ($d$DIC/$d$t and $d$DOC/$d$t, mg-C L$^{-1}$ d$^{-1}$) were considered in a two-layer conceptual
equation model as shown from **Equation 1** to **Equation 4**. The fluxes in the upper layer (from
the water surface to 2.5 m water depth) were calculated as follows:





$$V_U \frac{d\text{DIC}_U}{dt} = Q_U \text{DIC}_R - Q_{out} \text{DIC}_U - V_U \alpha_{PU} Chl_U + V_U \alpha_{MU} DOC_U + A_I w_I (\text{DIC}_L - \text{DIC}_U) \tag{1}$$

$$+ Q_L \text{DIC}_L - \frac{A_s F_{CO2}}{C_U} + Pa_U$$

$$V_U \frac{d\text{DOC}_U}{dt} = Q_U \text{DOC}_R - Q_{out} \text{DOC}_U - V_U \alpha_{MU} DOC_U + A_I w_I (\text{DOC}_L - \text{DOC}_U) \tag{2}$$

$$+ Q_L \text{DOC}_L + Pb_U$$

Those in the lower layer (from 2.5 to 4.0 m water depth) are calculated as follows:

$$V_L \frac{d\text{DIC}_L}{dt} = Q_L \text{DIC}_R - V_L \alpha_{PL} Chl_L + V_L \alpha_{ML} DOC_L + A_I w_I (\text{DIC}_U - \text{DIC}_L) - Q_L \text{DIC}_L \tag{3}$$

$$+ \frac{A_B B F_{DIC}}{C_U} + Pa_L$$

$$V_L \frac{d\text{DOC}_L}{dt} = Q_L \text{DOC}_R - V_L \alpha_{ML} DOC_L + A_I w_I (\text{DOC}_U - \text{DOC}_L) - Q_L \text{DOC}_L + Pb_L \tag{4}$$

$$V_{total} = V_U + V_L \tag{5}$$

$$Q_{in} = Q_U + Q_L \tag{6}$$

where, total lake volume ($V_{total}$, 53,544 m³) departs to the upper layer ($V_U$, 45,456 m³) and to
the lower layer ($V_L$, 8,808 m³) (**Equation 5**), and where lake surface area ($A_s$) is 36,000 m²
and the bottom of lake area ($A_B$) is 3,520 m². The interface is 2.5 m vertically, and the interface area
($A_I$) is 7,264 m² in YYL. The $C_U$ is a coefficient value (= 1,000) to establish a standard unit for
$F_{CO2}$ (mg-C m⁻² d⁻¹), considering the air–water $CO_2$ exchange by Fick's law as follows:

$$F_{CO2} = k_{CO2} \cdot K_H (pCO2_{water} - pCO2_{air}) \tag{7}$$

where $k_{CO2}$ is the gas transfer velocity from wind speed empirical equations (Cole and Caraco,
1998; Jähne et al., 1987; Smith, 1985; Wanninkhof, 1992). $K_H$ is Henry's coefficient calculated
by water temperature empirical equations (Plummer and Busenberg, 1982). $pCO2_{air}$ (µatm) is
the $CO_2$ partial pressure in the atmosphere by using air pressure data (Lin et al., 2021; Lin et al.,
2022), and the atmospheric $CO_2$ concentration is assumed to be 400 ppm. $pCO2_{water}$ (µatm) is
the $CO_2$ partial pressure at the water surface around 0.04 m water depth from water quality data
(temperature, pH, DIC concentration at water surface), and an empirical equation (Cai and
Wang, 1998) as followed by Lin et al. (2021). $F_{CO2}$ contributed approximately half of the net
ecosystem production (NEP) across the water surface to the atmosphere in YYL (Lin et al.,
2021). In addition, because sediment carbon may be an important flux into shallow subtropical



lakes, the sediment C flux ($BF_{DIC}$, $BF_{DOC}$, mg-C L$^{-1}$) in the lower layer should be considered
(Lin et al., 2022).

We assumed the river discharge and outflow discharge ($Q_{out}$, m$^3$ d$^{-1}$) are quasi–steady

state ($Q_{in} = Q_{out}$), dividing into upper discharge ($Q_U$, m$^3$ d$^{-1}$) and lower discharge ($Q_L$, m$^3$ d$^{-1}$)
(**Equation 6**). Lin et al. (2021) showed that the buoyancy frequencies in YYL are 0.011 ± 0.004
s$^{-1}$, 0.013 ± 0.004 s$^{-1}$, 0.006 ± 0.003 s$^{-1}$, and 0.007 ± 0.004 s$^{-1}$ from spring to winter, respectively,
inhibiting the vertical profile DIC mixed due to stratification. We estimated the percentages of
$Q_U$ and $Q_L$ based on the buoyancy frequency following Lin et al. (2020 and 2022). $Q_U$ values
were 75%, 80%, 45%, and 50% of $Q_{in}$ for spring to winter, respectively, and $Q_L$ values were
25%, 20%, 55%, and 50% of $Q_{in}$. The physical and biogeochemical regimes under climate
change remain uncertain, such as biological compositions, mixing regimes, morphometric
characteristics, air–water energy fluxes (evaporation and transpiration), and so on (Woolway et
al., 2020). To simulate extreme climate scenarios, we shifted the ratio of $Q_{in}$ for each season
and tested the river intrusion hypothesis (**Figure 1**). We established two extreme conditions,
labeled *Level 1* and *Level 2*. *Level 2* is the more extreme condition: $Q_U$ is 80% (spring), 85%
(summer), 50% (autumn), and 50% (winter) of $Q_{in}$; $Q_L$ is 20% (spring), 15% (summer), 50%
(autumn), and 50% (winter) of $Q_{in}$. *Level 1* is the condition between the present and the *Level 2*
condition: $Q_U$ is 77% (spring), 82% (summer), 47% (autumn), and 50% (winter) of $Q_{in}$; $Q_L$
is 23% (spring), 18% (summer), 53% (autumn), 50% (winter) of $Q_{in}$.

The contributions of photosynthesis production depended on the chlorophyll *a*

concentration ($Chl_U$, $Chl_L$, mg L$^{-1}$) and on the absorption coefficients in the upper layer ($\alpha_{PU}$, d$^{-1}$)
and lower layer ($\alpha_{PL}$, d$^{-1}$). The coefficients of DOC remineralization rates in the upper layer
($\alpha_{MU}$, d$^{-1}$) and lower layer ($\alpha_{ML}$, d$^{-1}$) were considered in the conceptual models as well. The $Pa_U$,
$Pa_L$, $Pb_U$, and $Pb_L$ are constants in the conceptual models. To obtain unknown values ($\alpha_{PU}$,
$\alpha_{MU}$, $\alpha_{PL}$, $\alpha_{MU}$, $\alpha_{ML}$, $w_I$, $BF_{DIC}$, $BF_{DOC}$, $Pa_U$, $Pa_L$, $Pb_U$, and $Pb_L$), we applied multiple
regression analysis. Additionally, these unknown values were tested by trial and error to obtain
the parameters of the *best-fit* condition (Nakayama et al., 2022), dividing the seasonal and
nonseasonal serranoids to learn the seasonal differences. The same parameters of the *best-fit*
condition were used to obtain the extreme conditions for *Level 1* and *Level 2*. We used the
coefficient of determination (R$^2$) and the Nash–Sutcliffe model efficiency coefficient (NSE)
(Nash and Sutcliffe, 1970) to quantify the performance of the equation model with DIC and DOC
sampling data (observation data) for each simulation.

### *2.3.3   NEP of DIC and DOC*

The net ecosystem production was defined as the difference between primary

production and ecological respiration due to photosynthesis and respiration via biota (Dodds



and Whiles, 2020). Because we assumed that the C fluxes were dependent on the river inflows
in YYL (**Figure 2**), we could estimate the NEP by end-member analysis using the C
concentration of the river inflow and outflow (Lin et al., 2021; Nakayama et al., 2020) by
following **Equation 8** to **Equation 11**. The upper layer NEP of DIC flux (mg C d$^{-1}$) was
obtained from **Equation 1** as follows:

$$
\begin{aligned}
\text{Upper NEP}_{\text{DIC}} &= C_U \alpha_{PU} Chl_U - C_U \alpha_{MU} DOC_U - C_U \frac{A_I w_I (\text{DIC}_L - \text{DIC}_U)}{V_U} - C_U \frac{Pa_U}{V_U} \\
&= C_U \frac{Q_U \text{DIC}_R + Q_L \text{DIC}_L - Q_{out} \text{DIC}_U}{V_U} - \frac{A_S}{V_U} F_{CO2} \\
&= C_U \frac{1}{t_{rU}} \left( \frac{Q_U}{Q_{in}} \text{DIC}_R + \frac{Q_L}{Q_{in}} \text{DIC}_L - \text{DIC}_U \right) - F_C \\
t_{rU} &= \frac{V_U}{Q_{in}}
\end{aligned}
\tag{8}
$$


The upper layer NEP of DOC flux (mg C d$^{-1}$) can be estimated from **Equation 2**:

$$
\begin{aligned}
\text{Upper NEP}_{\text{DOC}} &= C_U \alpha_{MU} DOC_U - C_U \frac{A_I w_I (\text{DOC}_L - \text{DOC}_U)}{V_U} - C_U \frac{Pb_U}{V_U} \\
&= C_U \frac{Q_U \text{DOC}_R + Q_L \text{DOC}_L - Q_{out} \text{DOC}_U}{V_U} \\
&= C_U \frac{1}{t_{rU}} \left( \frac{Q_U}{Q_{in}} \text{DOC}_R + \frac{Q_L}{Q_{in}} \text{DOC}_L - \text{DOC}_U \right)
\end{aligned}
\tag{9}
$$






The lower layer NEP of DIC flux (mg C d$^{-1}$) from **Equation 3**:

$$\text{Lower NEP}_{\text{DIC}} = C_U \alpha_{PL} Chl_L - C_U \alpha_{ML} DOC_L - C_U \frac{A_I w_I (\text{DIC}_U - \text{DIC}_L)}{V_L} - \frac{A_B BF_{DIC}}{V_L}$$

$$- C_U \frac{Pa_L}{V_L} = C_U \frac{Q_L (\text{DIC}_R - \text{DIC}_L)}{V_L} = C_U \frac{1}{t_{rL}} \frac{Q_L}{Q_{in}} (\text{DIC}_R - \text{DIC}_L) \qquad (10)$$

$$t_{rL} = \frac{V_L}{Q_{in}}$$


The lower layer NEP of DOC flux (mg C d$^{-1}$) from **Equation 4**:

$$\text{Lower NEP}_{\text{DOC}} = C_U \alpha_{ML} DOC_L - C_U \frac{A_I w_I (\text{DOC}_U - \text{DOC}_L)}{V_L} - \frac{A_B BF_{DOC}}{V_L} - C_U \frac{Pb_L}{V_L}$$

$$= C_U \frac{Q_L (\text{DOC}_R - \text{DOC}_L)}{V_L} = C_U \frac{1}{t_{rL}} \frac{Q_L}{Q_{in}} (\text{DOC}_R - \text{DOC}_L) \qquad (11)$$


Thus, the total NEP of DIC and DOC are:

$$\text{NEP}_{\text{DIC}} = \frac{V_U \text{Upper NEP}_{\text{DIC}} + V_L \text{Lower NEP}_{\text{DIC}}}{V_{total}} \qquad (12)$$

$$\text{NEP}_{\text{DOC}} = \frac{V_U \text{Upper NEP}_{\text{DOC}} + V_L \text{Lower NEP}_{\text{DOC}}}{V_{total}} \qquad (13)$$


where, $F_C$ is $\frac{A_S}{V_U} F_{CO2}$ and $t_{rU}$, $t_{rL}$ are residence times (d) in the upper and lower layers,
respectively. These parameters were used for the *best-fit* condition as shown in **Table 2**.



# 3.    Results

## *3.1    DIC, DOC, and Chl. a concentrations in typhoon and non-typhoon years*

Our results demonstrated that there were no significant differences in DIC, DOC, and Chl. *a* concentration between the layers in the typhoon years 2015 and 2016; on the other hand, all these parameters differed significantly between the layers in the typhoon years 2017 and 2018 (**Figure 3**). The average $DIC_U$ was 1.23 mg-C $L^{-1}$, and $DIC_L$ was 3.66 mg-C $L^{-1}$; the average $DOC_U$ was 5.87 mg-C $L^{-1}$, and $DOC_L$ was 8.02 mg-C $L^{-1}$; and the $Chl_U$ and $Chl_L$ were 18.5 µg-C $L^{-1}$ 2.13 µg-C $L^{-1}$, respectively (**Figure 3**). However, t-test results showed no significant differences in DIC, DOC, and Chl. *a* concentrations (*p*-values ≥ 0.05 that no significant differences in DIC data among seasons in the typhoon years) (**Figure 4 a**). On the other hand, the DOC concentration showed significant differences between seasons in the typhoon years (**Figure 4 c-d**). No significant differences between $Chl_U$ and $Chl_L$ were observed among the seasons (**Figure 4 e-f**). However, the standard deviations (SD) of DIC and DOC were higher in summer and autumn (**Figure 4**) due to terrestrial C loading (Chiu et al., 2020). In summer, the SD values of $DIC_U$ and $DOC_U$ were 3.51 mg-C $L^{-1}$ and 3.69 mg-C $L^{-1}$, respectively (**Figure 4 a, c, e**). In autumn, $DIC_L$ and $DOC_L$ had the highest SD (4.06 and 4.17 mg-C $L^{-1}$, respectively) (**Figure 4 b, d**). Notably, the maximums of $DIC_U$ and $DOC_U$ were 7.06 and 15.6 mg-C $L^{-1}$ and those of $DIC_L$ and $DOC_L$ were 10.9 and 19.8 mg-C $L^{-1}$, respectively, in the typhoon years (**Figure 4 a-d**).

Positive Pearson correlations of 0.45 to 0.80 were observed between the DOC and DIC in the typhoon years (**Figure 5 a**). In the non-typhoon years, the upper layer $DIC_L$ was the only variable negatively correlated with DOC in the upper and lower layers (**Figure 5 b**). The lower layer DIC was positively correlated with the $Chl_L$ due to the abundant respiration in the lower layer (**Figure 5**).



### *3.2    Performance of conceptual two-layer DIC and DOC models*
The results for the typhoon years demonstrated that the most of the seasonal scenarios
were better fitting than the nonseasonal scenarios (**Figure 6**). Under the seasonal scenarios, the
$DIC_U$ was around 1.5 to 5.0 mg-C L$^{-1}$ (**Figure 6 a-b**) and $DIC_L$ was around 5.0 mg-C L$^{-1}$ stably
(**Figure 6 d**). However, the NSE of $DIC_L$ was 0.73 under the nonseasonal scenarios, which was
higher than seasonal scenarios (NSE = 0.71) (**Table 2**), because $DIC_L$ was elevated dramatically,
by 40 mg-C L$^{-1}$, under the nonseasonal scenarios during the 2016 typhoon period (**Figure 6 c**). In
the non-typhoon years (2017-2018), the *best-fit* values of $DIC_U$ and $DIC_L$ did not differ
significantly between the seasonal and nonseasonal scenarios (R$^2$ and NSE were around 0.40 and
0.70, respectively). These results demonstrated that $DIC_U$ and $DIC_L$ in the typhoon years must
use the seasonal scenarios, whereas in the non-typhoon years they should use the nonseasonal
scenarios. On the other hand, the DOC under the seasonal scenarios fit our observation data
perfectly (R$^2$ = 0.91, 0.46 and NSE = 0.95, 0.73 for $DOC_U$, $DOC_L$, respectively) (**Figure 6 e-h**,
**Table 2**). Thus, the results suggested that the $DOC_U$ and $DOC_L$ must use the seasonal scenarios
in both the typhoon and non-typhoon years.
As shown in **Table 2**, the parameters for the conceptual two-layer DIC and DOC models
showed different regimes between the typhoon and non-typhoon years. In the typhoon years, the
photosynthesis absorption rates ($\alpha_{PU}$, $\alpha_{PL}$) were negative (photosynthesis < respiration) for each
season. YYL was a C source due to a large allochthonous C loading during typhoons; the
respiration was elevated by around 30- to 150-fold from summer to autumn. On the other hand,
the transportation rates ($w_I$, $w_{IL}$) were higher in autumn than in the other seasons (**Table 2**) due
to weak stratification and large C loading during typhoons. Additionally, the higher
remineralization rates during typhoon disturbances from summer to autumn resulted in positive
$\alpha_{MU}$ and $\alpha_{ML}$. In the non-typhoon years, the remineralization rates were negative (**Table 2**).
Thus, the results suggested that the conceptual two-layer C models may be reasonably to fit the
observation data.





**_3.3      Interannual and seasonal NEP in YYL_**

The typhoon disturbances in summer and autumn played an important role in

promoting the C released by YYL (**Table 3**). Overall, YYL released 245 mg-C d$^{-1}$ of DIC and
415 mg-C d$^{-1}$ of DOC during the typhoon years; during the non-typhoon years, it released 51.7
mg-C d$^{-1}$ of DIC and 22.8 mg-C d$^{-1}$ of DOC (**Table 3**). The average $F_C$ was one to two times
larger than NEP, and 219 and 133 mg-C d$^{-1}$ were released from YYL into the atmosphere in the
typhoon and non-typhoon years, respectively (**Table 3**). In summer, the upper layer DIC and
DOC consumed approximately 3.7 times more DIC in the typhoon years than in the non-
typhoon years (**Table 3**). In autumn, 216 mg-C d$^{-1}$ of upper layer DIC was released; however,
46.1 mg-C d$^{-1}$of upper layer DOC was produced in the typhoon years. The upper layer $\text{NEP}_{\text{DIC}}$
was negative in autumn of the typhoon years, when 268 mg-C d$^{-1}$ more $F_C$ was released
compared to the non-typhoon years. In addition, the lower layer was most released of C into the
outflow; however, the NEP in the lower layer was more than twice as high in summer than in
autumn in the typhoon years (**Table 3**). The average of total $\text{NEP}_{\text{DIC}}$ was 3.14 times more
released C in the typhoon than in the non-typhoon years; The average of total $\text{NEP}_{\text{DOC}}$ was
increased 62.3 mg-C d$^{-1}$ of DOC between the typhoon years and non-typhoon years due to the
over ten-times higher NEP in the upper layer (**Table 3**).

The ratios of DIC and DOC concentrations reveal the magnitudes of allochthonous

DOC loading into YYL (Shih et al., 2019; Walvoord and Striegl, 2007), and the upper and lower
layers show different patterns. In the typhoon years, the upper layer ratios decreased (higher
DOC loading) from summer to autumn, whereas in the lower layer, DIC:DOC decreased from
autumn to winter. In the non-typhoon years, the autumn DIC:DOC was the lowest, around 0.216
to 0.351 (**Table 3**).





### *3.4      Interannual responses of DIC and DOC to typhoons*

We simulated the responses of DIC and DOC flux to typhoons by using conceptual

two-layer C models. The results showed that the DIC was more sensitive to typhoon
disturbances than DOC under scenarios of *Level 1* and *Level 2* (**Figure 7-9**). Overall, the C level
declined in the upper layers but increased in the lower layers (**Figure 7**). DIC and DOC in the
upper layer tended to decline from 1.0 (*Level 1*) to 2.0 mg-C L$^{-1}$ (*Level 2*) (**Figure 7 a, c**); at the
same time, they increased to 10.0 and 20.0 mg-C L$^{-1}$ in the lower layer under *Level 1* and *Level*
*2*, respectively (**Figure 7 b, d**).

The DIC concentration in the upper layer was significantly lower in typhoon than non-

typhoon years during spring and autumn under *Level 2* (**Figure 8 a-c**). Under the *best-fit* and
*Level 1* conditions, DIC concentrations decreased significantly from winter to spring (**Figure 8**
**c-d**). The lower layer DIC values under the *best-fit* and *Level 1* conditions differed significantly
between the typhoon and non-typhoon years (**Figure 8 e-h**). The lower layer DIC under *Level 2*
differed significantly from winter to spring only (**Figure 8 e, h**). On the other hand, upper layer
DOC showed significant typhoon responses for each condition from winter to spring (**Figure 9**
**a, d**). The upper layer DOC tended to differ more significantly under the extreme climates from
summer to autumn (**Figure 9 b-c**). The lower layer DOC showed different typhoon responses
between spring and the other seasons (**Figure 9 e-h**).



## 4.      Discussion

Annual total precipitation was 40% higher in typhoon years than in non-typhoon years (**Table 1**).Water retention and typhoon-induced upwelling control the dynamics of DIC and DOC during summer and autumn (Chiu et al., 2020; Jones et al., 2009; Tsai et al., 2008; Tsai et al., 2011). Typhoon-induced upwelling affects water quality data differently between upper and lower layers (**Figure 3**). DIC, DOC, and Chl. *a* concentrations differ significantly between upper and lower layers in the typhoon years (**Figure 3**) due to thermal stratification (Chiu et al., 2020; Lin et al., 2022; Tsai et al., 2008; Tsai et al., 2011). In addition, the abundance of organisms leads to intensive respirations in the lower layers during the non-typhoon period; for example, an anoxic condition at the hypolimnion may affect C mineralization and remineralization rates in non-typhoon years (Carey et al., 2022; Chiu et al., 2020; Lin et al., 2022; Shade et al., 2010; Shade et al., 2011). Therefore, these physical and biogeochemical processes might describe different patterns between the upper and lower layers, as revealed by Pearson correlations (**Figure 5**).

Thermal stratification and allochthonous C loading may drive the responses of NEP to typhoons in YYL. In the typhoon years, the absolute values of NEP were higher than in the non-typhoon years (**Table 3**). We found that precipitation from typhoons loaded large amounts of allochthonous C into YYL during summer and autumn, which might describe the higher NEP in autumn compared to other seasons (**Table 3**). Typhoons dramatically changed the seasonal and interannual patterns of DIC fluxes due to river intrusion (**Figure 7 a-b; Figure 8**), which corresponds to our hypothesis (**Figure 1**) and to the results of previous studies (Chiu et al., 2020; Lin et al., 2021; Lin et al., 2022). In summer, the spatial differences in DIC and DOC between layers were inhibited due to strong thermal stratification, describing the positive upper NEP and lower negative NEP (Lin et al., 2021).The thermal stratification and anoxic condition may have been controlled by the seasonal and interannual patterns of DIC and DOC fluxes in the non-typhoon years (**Tables 2-3; Figure 6**). Additionally, because of the absence of typhoon-induced mixing and allochthonous C loading, the absolute values of total NEP in the non-typhoon years were less than those the non-typhoon years (**Table 3**). These results suggested that the allochthonous C loading was the most crucial factor for DIC and DOC fluxes in the typhoon years; on the other hand, the transportation rate shaped the seasonal C due to thermal stratification in the non-typhoon years.

With the conceptual two-layer C models (**Table 2**), photosynthesis absorption ($\alpha_{PU}$, $\alpha_{PL}$), remineralization ($\alpha_{MU}$, $\alpha_{ML}$), and transportation ($w_I$, $w_{IL}$) well represented the seasonal variations of DIC and DOC data. These parameters of the conceptual two-layer C models appeared in reasonable patterns (**Table 2**). The higher remineralization and photosynthesis rates resulted in higher absolute values of NEP in the autumn of the typhoon years (**Tables 2-3**). In





the non-typhoon years, the photosynthesis rates contributed to the total NEP (**Tables 2-3**).
Moreover, without the typhoon-induced mixing and refreshing of the water column, anoxic
conditions may occur (Carey et al., 2022; Vachon et al., 2021), which could result in negative
remineralization rates in non-typhoon years. Thus, the conceptual two-layer C models well
characterizes the seasonal and interannual responses of DIC and DOC fluxes to typhoons in
YYL.

Under extreme weather conditions, *Level 2* usually shifted to different typhoon

responses for each season (**Figures 8-9**) due to extreme river intrusions. DIC changes more
significantly than DOC under *Level 1* and *Level 2* (**Figures 7-9**), because the photosynthesis,
transportation, and remineralization rates may crucially affect the seasonal and interannual
patterns of DOC as well (**Figure 1**). Moreover, we compared the NEP with different model
conditions as shown in **Figure 10**, demonstrating that the responses of $NEP_{DIC}$ to typhoons
differed dramatically between *Level 1* and *Level 2* (**Figure 10 a-c**); especially, the Upper
$NEP_{DIC}$ released more C in the typhoon years and absorbed more C in the non-typhoon years
than *Obs* (**Figure 10 a**). Not only were the absolute values of $NEP_{DIC}$ over 3 times higher in the
typhoon than the non-typhoon years (**Table 3**), but SD was higher in the typhoon years as well
(**Figure 10**). However, $NEP_{DOC}$ changed less under *Level 1* and *Level 2* (**Figure 10 d-f**), a
finding that is consistent with our continuous DOC data (**Figure 7 c-d**). Processes such as
respiration, mineralization, and sediment burial may impact DOC fluxes (Bartosiewicz et al.,
2015; Hanson et al., 2015; Maranger et al., 2018). To our knowledge, bio-photochemical
mineralization and degradation may play a key role in shaping C fluxes because colored DOC
reduced ultraviolet radiation (UVR) and active photosynthetic radiation (PAR) (Allesson et al.,
2021; Chiu et al., 2020; Schindler et al., 1996; Scully et al., 1996; Williamson et al., 1999).
Thus, we suggest that photo-biochemical processes (such as the photo-mineralization) should be
considered in the upper layer in order to clarify and validate the responses of the total C fluxes
under extreme climates in a two-layer stratified lake.





## 5.    Conclusions

Our conceptual two-layer C model revealed that allochthonous and autochthonous processes both accounted for C flux responses to typhoon disturbances on seasonal and interannual scales by applying our proposed two-layer conceptual C model. Without typhoons, the strength of thermal stratification were the primary determinants the seasonal and interannual patterns of DIC and DOC concentrations data and NEP. Typhoon-induced upwelling and loading facilitated 102.2 mg-DIC d$^{-1}$ and 62.3 mg-DOC d$^{-1}$ in YYL, respectively (**Table 3**). We successfully developed two-layer conceptual C models to obtain continuous DIC and DOC data in YYL and to simulate extreme conditions. The changes in seasonal river intrusion regimes in YYL resulted in a 3-fold higher total NEP$_{DIC}$ in the typhoon years than in the non-typhoon years. However, our model should be improved under extreme climate scenarios by considering other autochthonous processes, such as sediment burial, photo-biochemical processes, and anoxic conditions. The present results suggest that physical processes (river intrusion and vertical transportation) and biogeochemical processes (mineralization, photosynthesis, and respiration) in a subtropical small lake accounted for the C flux responses to typhoons on seasonal and interannual time scales.



**Competing interests**

The authors have no conflicts of interest to report.


**Acknowledgements**

The authors thank YS Hsueh, LC Jiang, and TY Chen for their help with water sample

collection and chemistry analysis. This work was supported by the Japan Society for the
Promotion of Science (JSPS) under grant nos. 22H05726, 22H01601, and 18KK0119 for K
Nakayama; and by the Academia Sinica, Taiwan (AS-103-TP-B15), Ministry of Science and
Technology, Taiwan (MOST 106-2621-M-239-001, MOST 107-2621-M-239-001, MOST 108-
2621-M-239-001) for CY Chiu and JW Tsai. This study benefited from participation in the
Global Lakes Ecological Observatory Network (GLEON).

Hao-Chi Lin: Conceptualization, Methodology, Investigation, Formal analysis,

Writing – original draft. Keisuke Nakayama: Methodology, Supervision, Writing – review &
editing, Conceptualization. Jeng-Wei Tsai: Investigation, Funding acquisition, Writing –review
& editing. Chih-Yu Chiu: Funding acquisition, Writing – review & editing.



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



**Table 1.** Comparison of rainfall and hydrological records between typhoon and non-typhoon
years at Yuan–Yang Lake.

| Records | Typhoon years | Non-typhoon years |
|---|---|---|
| Time period (year) | 2015-2016 | 2017-2018 |
| Total precipitation (mm) | 6,332 | 3,795 |
| Total typhoon rainfall (mm) | 2,254 | 0 |
| Average water depth (m ± SD) | 4.54 ± 1.7 | 4.51 ± 1.5 |
| Average river discharge ($m^3$ $d^{-1}$) | 3,717 | 2,943 |
| Transparency (Secchi disc depth, m ± SD) | 1.58 ± 0.45 | 1.38 ± 0.28 |




**Table 2.** *Best-fit* parameters of DIC and DOC two-layer conceptual model from 2015 to 2018.

| | 2015–2016 Typhoon years | | | | 2017–2018 Non-typhoon years | | | | 2015–2016 Typhoon years | 2017–2018 Non-typhoon years |
|---|---|---|---|---|---|---|---|---|---|---|
| | *Spring* | *Summer* | *Autumn* | *Winter* | *Spring* | *Summer* | *Autumn* | *Winter* | *Non-seasonal* | *Non-seasonal* |
| *Upper layer* | | | | | | | | | | |
| $F_{CO2}$ (mg-C m$^2$ d$^{-1}$) | 291 | 245 | 422 | 127 | 231 | 143 | 104 | 175 | 276 | 163 |
| $\alpha_{PU}$ (d$^{-1}$) | -1.20 | -33.1 | -183.5 | -29.1 | 8.0 | 6.0 | 30.0 | 7.77 | -22.0 | 8.0 |
| $\alpha_{MU}$ (d$^{-1}$) | -0.0227 | 0.0203 | 0.08 | -0.031 | -0.01 | -0.039 | -0.033 | -0.195 | -0.035 | -0.0238 |
| $w_I$ (d$^{-1}$) | 0.230 | 0.172 | 1.38 | 0.30 | 0.10 | 0.0478 | 0.120 | 0.180 | 0.159 | 0.107 |
| $Pa_U$ (d$^{-1}$) | 12560 | -1317 | -23750 | 9597 | 9880 | 14000 | 17600 | 10100 | 4457 | 12420 |
| $Pb_U$ (d$^{-1}$) | -21930 | 9461 | -42130 | -17070 | -3630 | -1251 | -20820 | -9289 | -12760 | -9119 |
| $d\mathrm{DIC}_U$ (R$^2$, NSE) | 0.305, 0.614 | | | | | | | | 0.072, 0.299 | 0.403, 0.650 |
| $d\mathrm{DOC}_U$ (R$^2$, NSE) | 0.909, 0.953 | | | | | | | | 0.242, 0.569 | 0.320, 0.918 |
| *Lower layer* | | | | | | | | | | |
| $\alpha_{PL}$ (d$^{-1}$) | -0.627 | -22.1 | 15.0 | -0.878 | 1.49 | -6.87 | 6.0 | -16.6 | -21.11 | 2.0 |
| $\alpha_{ML}$ (d$^{-1}$) | -0.025 | 0.123 | 0.0755 | 0.00973 | -0.010 | -0.0376 | -0.04 | -0.048 | 0.123 | -0.019 |
| $w_{IL}$ (d$^{-1}$) | 0.205 | 0.187 | 0.540 | 0.286 | 0.112 | 0.055 | 0.298 | 0.166 | 0.0868 | 0.176 |
| $Pa_L$ (d$^{-1}$) | 100 | -5662 | -10500 | -1013 | 151.6 | 2032 | 1216 | 909 | -5662 | -40.5 |
| $Pb_L$ (d$^{-1}$) | -6012 | -7395 | -53940 | -9639 | -1338 | -6296 | -19470 | -8748 | -12240 | -9919 |
| $BF_{DIC}$, $BF_{DOC}$ (mg-C L$^{-1}$) | 0.04, 0.00 | | | | | | | | | |
| $d\mathrm{DIC}_L$ (R$^2$, NSE) | 0.452, 0.707 | | | | | | | | 0.192, 0.306 | 0.440, 0.731 |
| $d\mathrm{DOC}_L$ (R$^2$, NSE) | 0.460, 0.728 | | | | | | | | 0.234, 0.338 | 0.128, 0.525 |






**Table 3.** Seasonal averages of C fluxes (mg-C d⁻¹) for each season in YYL. Positive values are
shown in the C sink (*black*), and negative ones show the values after C was released (*red*).

| | | Flux (mg C d⁻¹) | | | $DIC_U$ / $DOC_U$ | $DIC_L$ / $DOC_L$ | Total (mg C d⁻¹) | |
|---|---|---|---|---|---|---|---|---|
| | | $F_C$ | Upper NEP | Lower NEP | | | $NEP_{DIC}$ | $NEP_{DOC}$ |
| *Typhoon years* | **Average:** | **-219** | - | - | - | - | **-150** | **-9.69** |
| Spring | DIC | -231 | -243 | -45.2 | 0.658 | 0.568 | -210 | 62.1 |
| | DOC | - | 70.8 | 17.2 | | | | |
| Summer | DIC | -194 | 29.1 | -313 | 0.193 | 0.511 | -26.4 | 18.8 |
| | DOC | - | 118 | -495 | | | | |
| Autumn | DIC | -351 | -216 | -659 | 0.349 | 0.475 | -288 | -151 |
| | DOC | - | 46.1 | -1167 | | | | |
| Winter | DIC | -100 | -96.4 | 36.5 | 0.442 | 0.372 | -74.8 | 31.2 |
| | DOC | - | 40.5 | -16.9 | | | | |
| *Non-typhoon years* | **Average:** | **-133** | - | - | - | - | **-47.8** | **52.6** |
| Spring | DIC | -129 | -180 | -94.9 | 0.524 | 0.634 | -166 | -7.06 |
| | DOC | - | 21.4 | -67.1 | | | | |
| Summer | DIC | -183 | 5.80 | -58.1 | 0.260 | 0.423 | -4.57 | 73.8 |
| | DOC | - | 115 | -140 | | | | |
| Autumn | DIC | -82.6 | 95.0 | 35.9 | 0.216 | 0.351 | 85.5 | 95.9 |
| | DOC | - | 168 | -272 | | | | |
| Winter | DIC | -138 | -128 | 6.04 | 0.449 | 0.436 | -106 | 33.7 |
| | DOC | - | 34.0 | 32.1 | | | | |





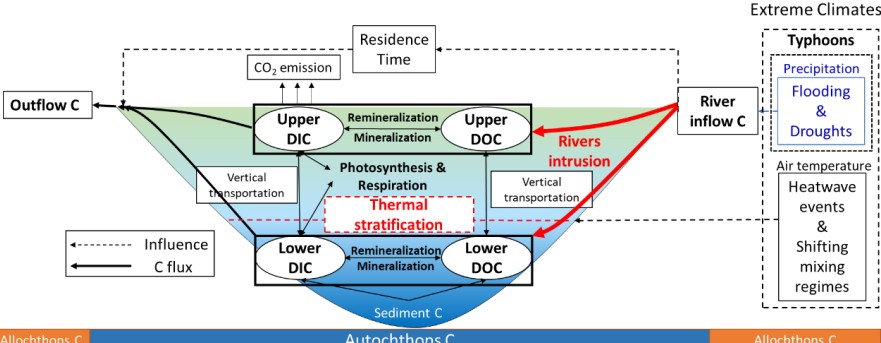

*Figure 1.* Conceptual diagram of river intrusion (*red arrows*) and thermal stratification (*red dashed line*) dominant responses of DIC and DOC in a subtropical two-layer stratified lake under extreme climates.



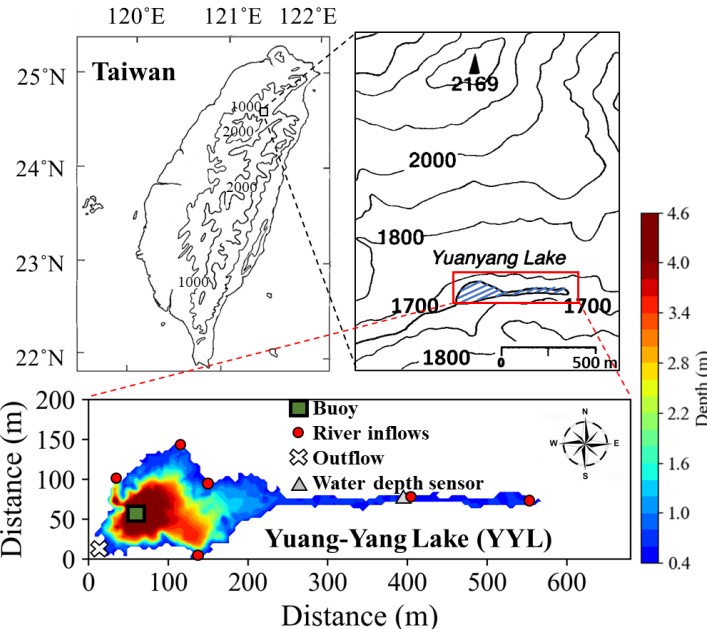

680

**Figure 2**. Sampling locations and bathymetry maps of Yuan–Yang Lake (YYL). The
*dark green rectangle* shows the buoy station, which is at the deepest site of the lake. The
*red points* and *white cross* show the river mouths of the inflows and outflow,
respectively. The *gray triangle* shows the location of the water depth sensor.

685

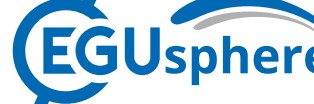

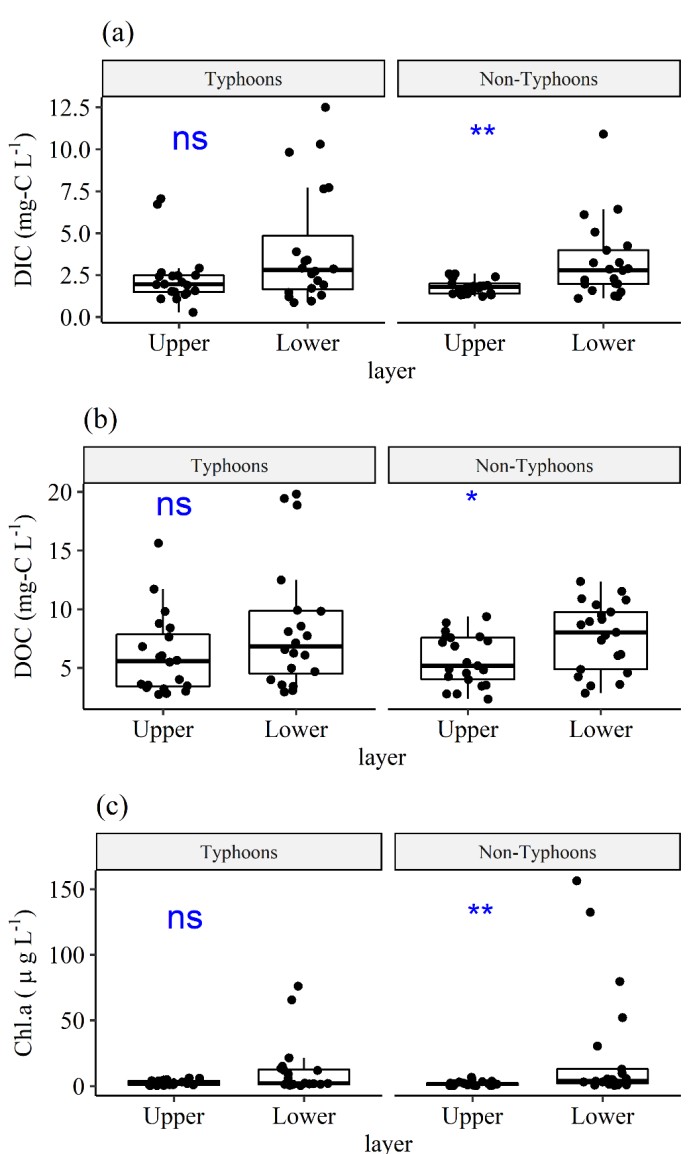

**Figure 3**. Comparisons of **(a)** DIC, **(b)** DOC, and **(c)** Chl. *a* between upper ($DIC_U$, $DOC_U$, $Chl_U$) and lower ($DIC_L$, $DOC_L$, $Chl_L$) layers, grouped by typhoon and non-typhoon years. The *bullet points* show the water sampling data. We used a t-test to obtain *p*-values. The **ns** show *p*-values ≥ 0.05, * show *p*-values from 0.05 to 0.01, and ** show *p*-values from 0.01 to 0.001 by t-test.



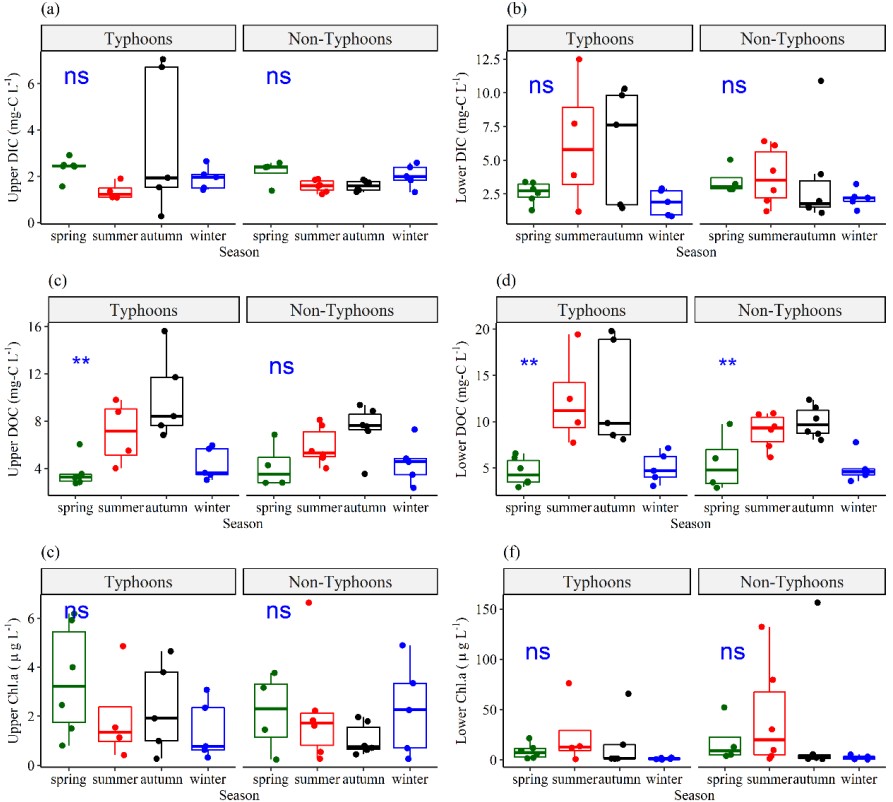

692

**Figure 4.** Seasonal variations of (**a**) upper layer DIC ($DIC_U$), (**b**) lower layer DIC
($DIC_L$), (**c**) upper layer DOC ($DOC_U$), (**d**) lower layer DOC ($DOC_L$), (**e**) upper layer Chl.
*a* ($Chl_U$), (**f**) lower layer Chl. *a* ($Chl_L$) grouped by typhoon and non-typhoon years. The
*bullet points* show water sampling data. To know the seasonality, we used one-way
ANOVA to obtain the p-values. The **ns** show p-values ≥ 0.05, **\*** show p-values from
0.05 to 0.01, and **\*\*** show *p*-values are from 0.01 to 0.001.

699




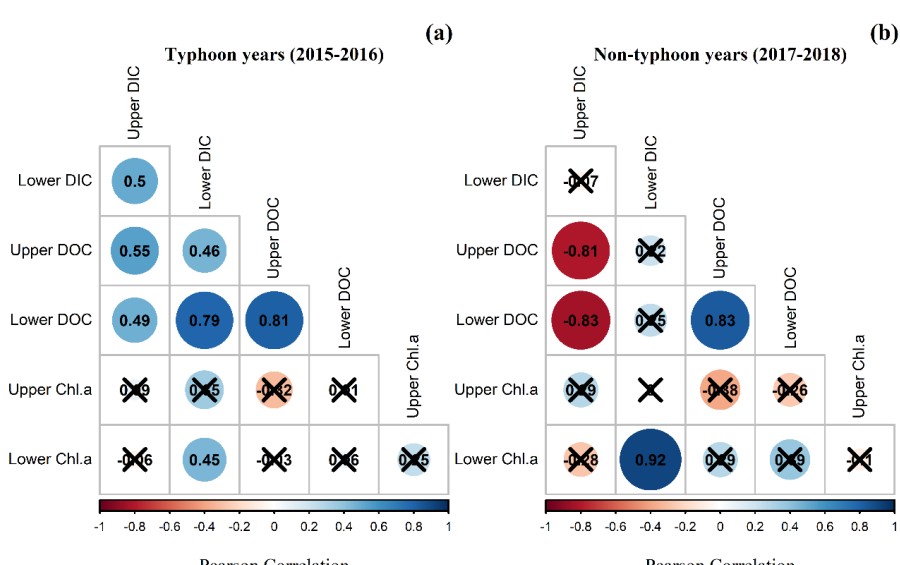

**Figure 5.** Pearson correlation coefficients of DIC, DOC, Chl. a concentration at upper
layer and lower layer DIC ($DIC_U$, $DIC_L$), DOC ($DOC_U$, $DOC_L$), Chl. $a$ ($Chl_U$, $Chl_L$)
during **(a)** typhoon years and **(b)** non-typhoon years. The *black-crosse*s show
insignificant values (*p*-values are > 0.05).

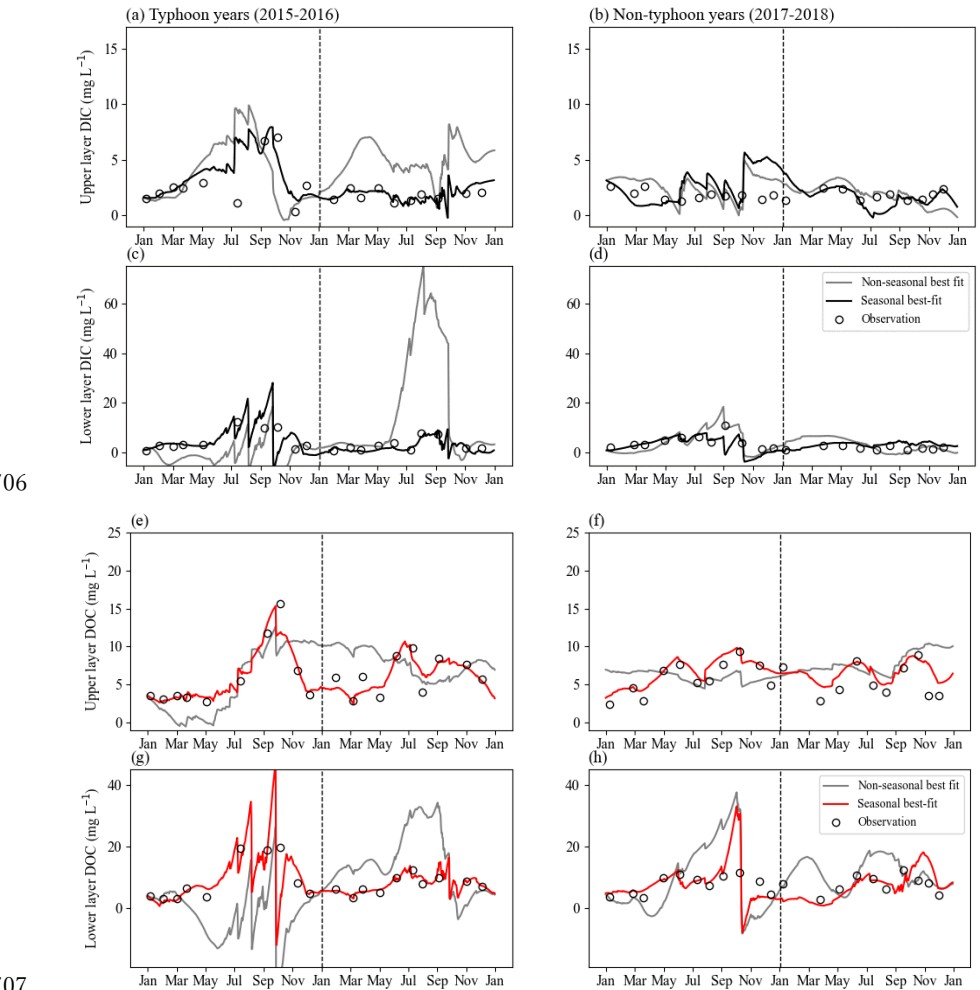

706

707

**Figure 6.** Continuous daily DIC and DOC data at **(a, b, e, f)** upper layer ($DIC_U$, $DOC_U$)
and **(c, d, g, h)** lower layer ($DIC_L$, $DOC_L$) by using conceptual equations models. The
gray lines show the original data, the *blue lines* show the nonseasonal data, the *black lines*
show the *best fit* for DIC, the *red lines* show the *best-fit* for DOC (**Table 2**), and the *empty
dots* show water sampling (observation) data for each month.

713

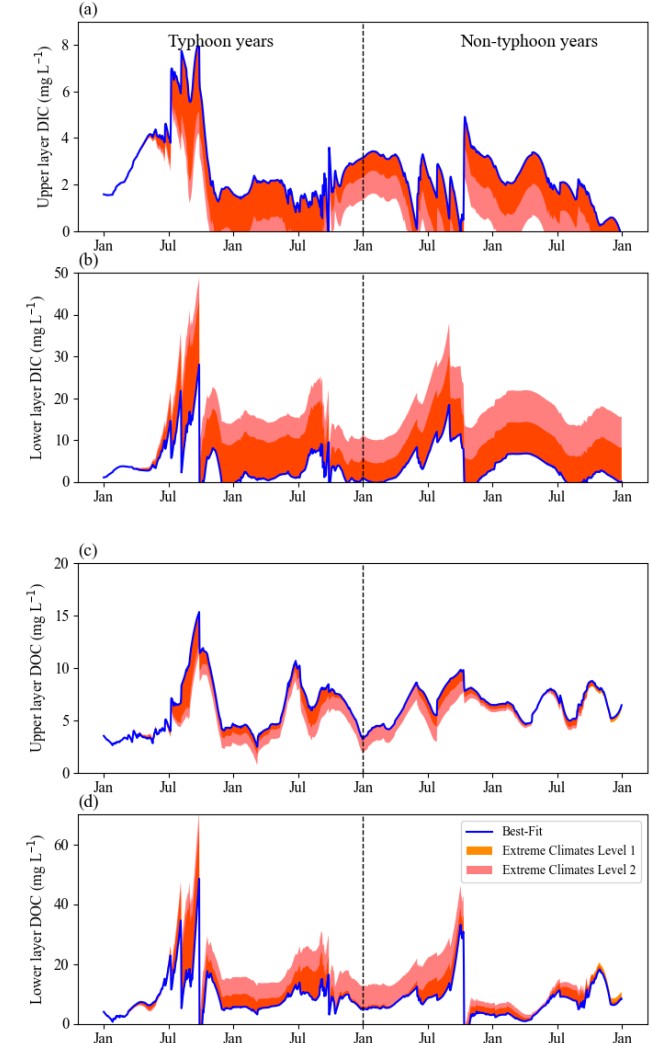

714

**Figure 7.** Continuous daily DIC and DOC data at (**a, c**) upper layer ($DIC_U$, $DOC_U$) and
(**b, d**) lower layer ($DIC_L$, $DOC_L$) by using the conceptual equation model under extreme
climates from 2015 to 2018. *Blue lines* are original best-fit data as in **Figure 4**, in which
the parameters of the DIC model in non-typhoon years are under the nonseasonal
scenario and the others are under the seasonal scenario as in **Table 2**. *Orange regions*
show *Level 1*; *pink regions* show *Level 2*.




721



**Figure 8.** Seasonal responses of continuous **(a-d)** upper layer DIC and **(e-h)** lower layer DIC (mg-C L$^{-1}$) between typhoon (*Typhoon*) and non-typhoon (*Non*) years for each season as in **Figure 8**. *Fit* (*blue boxes*) condition shows the best-fit data by using the conceptual two-layer C model; *Lv1* (*yellow boxes*) and *Lv2* (*red boxes*) show the extreme climates. The *empty dots* show the continuous DIC and DOC data. The **ns** show $p$-values $\geq 0.05$, **\*** show $p$-values from 0.05 to 0.01, **\*\*** show $p$-values from 0.01 to 0.001; **\*\*\*\*** show $p$-values less than 0.0001 by using a t-test.



729





**Figure 9.** Seasonal responses of **(a-d)** upper layer DOC and **(e-h)** lower layer DOC (mg-C L$^{-1}$) between typhoon (*Typhoon*) and non-typhoon (*Non*) years for each season as in **Figure 8**. *Fit* (*blue boxes*) condition shows the best-fit data by using the conceptual two-layer C model; *Lv1* (*yellow boxes*) and *Lv2* (*red boxes*) show the extreme climates. *Empty dots* show the continuous DIC and DOC data. The **ns** show *p*-values $\geq 0.05$, **\*** show *p*-values from 0.05 to 0.01, **\*\*** show *p*-values from 0.01 to 0.001; **\*\*\*\*** show *p*-values less than 0.0001 by using a t-test.

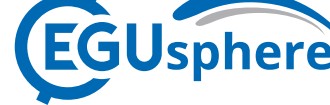

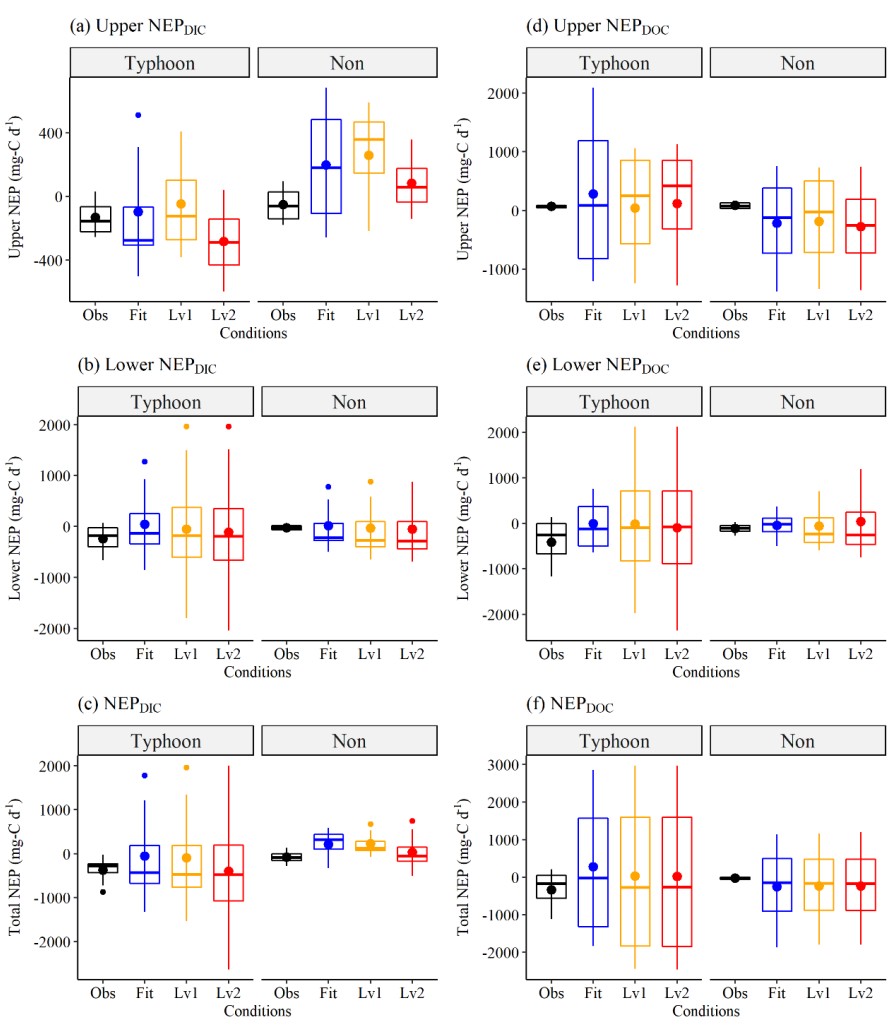

**Figure 10.** Interannual (a) Upper $NEP_{DIC}$, (b) Lower $NEP_{DIC}$, (c) $NEP_{DIC}$, (d) Upper $NEP_{DOC}$, (e) Lower $NEP_{DOC}$, and (f) $NEP_{DOC}$ flux (mg-C d$^{-1}$) grouped by typhoon and non-typhoon years. *Obs* condition (*black boxes*) show the observation data as in **Figure 6**; *Fit* condition (*blue- boxes*) show the best-fit data by using the conceptual two-layer C model as in **Figure 6**; *Level 1* (*yellow boxes*) and *Level 2* (*red boxes*) show the extreme scenarios as in **Figure 7**.