# Peer review of "Conceptual models of dissolved carbon fluxes in a two-layer"

_EGUsphere, 2022_

## Author Comment (AC1)

*In this paper a two-layer conceptual C models was developed in a small subtropical lake to explore how the DIC and DOC fluxes respond to typhoon disturbances on seasonal and interannual time scales. Monthly field samplings were conducted to measure DIC, DOC, and chlorophyll a concentrations to compare the temporal patterns of fluxes between typhoon years and non-typhoon years. It is an interesting study, and the manuscript need to be revised.*

Response: Thanks for your comments. The manuscript has been revised, taking into account your comments below.

*(1) Line 176-179, "where, total lake volume ($V_{total}$, 53,544 $m^3$) departs to the upper layer ($V_U$, 45,456 $m^3$) and to the lower layer ($V_L$, 8,808 $m^3$) (**Equation 5**), and where lake surface area ($A_S$) is 36,000 $m^2$ and the bottom of lake area ($A_B$) is 3,520 $m^2$. The interface is 2.5 m vertically, and the interface area ($A_I$) is 7,264 $m^2$ in YYL." The volume of upper layer and lower layer may change in time of different month, it is not a constant number and better to give the explanations.*

Response: Thanks for your comment. We have added, "The water depth is steady and changes. However, the change in water depth ranges from 4.56 to 4.66 m during typhoon period. Therefore, we can assume that the changes inf lake volumes and areas were negligible." in the manuscript.

*(2) Line 225-226 "2.3.3 NEP of DIC and DOC, the net ecosystem production was defined as the difference between primary production and ecological respiration due to photosynthesis and respiration via biota". The net ecosystem production has close relationship with water temperature and solar radiation in each month, especially in non-typhoon years. So, the discussions on the effects on NEP by temperature and solar radiation may be important.*

Response: Thanks for your suggestion. We have added a paragraph about the seasonal change of DIC and DOC fluxes in the discussion.

*(3) In the discussion, the $CO_2$ emission flux in different month for the small subtropical lake may be more interesting.*

Response: Thanks for your comment. We have added some sentences about the seasonal change of $CO_2$ emission in the discussion.

---

## Author Comment (AC2)

*Overall, this is an interesting study that measured DOC, DIC and Chl a in a small lake monthly for two years of contrasting precipitation. Water inputs and outputs were estimated and the observed concentrations were compared to a model predicting daily concentrations and fluxes for a two year period. The model output was further explored by simulating results from two climate scenarios that altered the water distribution terms in the model. The motivation and objectives to understanding precipitation-driven influence on lake carbon cycling align with the journal scope and the results are likely of interest to readers if the work can be more clearly communicated. While the paper is generally well-organized, the methods and results lack detail and clarity. This manuscript requires further careful editing for English grammar and spelling throughout. I commend the authors for their efforts in merging field data collection with modeling on this important topic and I hope my comments below are constructive.*

Response: I appreciate your positive and constructive comments. We have checked the manuscript thoroughly and completed an English proofreading. The manuscript has been revised, taking into account your comments as below.

**General comments**

*(1) No data availability statement was included.*

Response: We have added the data availability section,

"The data that support the findings of this study are adopted from our previous works, including Chiu et al. 2020, Lin et al. 2021, and Lin et al. 2022."

*(2) Given the small size and shallow depth of this lake, does a single volume model predict average DIC, DOC, Chl a, and $CO_2$ evasion dynamics just as well as a two-layer model?*

Response:   Because the thermal stratification was a vital process that controls the vertical profile of carbon concentration in YYL (Lin et al. 2021), this suggests that the two-layer system is more reasonable for charactering the DIC and DOC dynamics within the lake..

References:
Lin, H.-C., Chiu, C.-Y., Tsai, J.-W., Liu, W.-C., Tada, K., and Nakayama, K.: Influence of Thermal Stratification on Seasonal Net Ecosystem Production and Dissolved Inorganic Carbon in a Shallow Subtropical Lake, J. Geophys. Res.

Biogeosci., 126, https://doi.org/10.1029/2020JG005907, 2021.

Nakayama, K., Kawahara, Y., Kurimoto, Y., Tada, K., Lin, H.-C., Hung, M.-C., Hsueh, M.-L., and Tsai, J.-W.: Effects of oyster aquaculture on carbon capture and removal in a tropical mangrove lagoon in southwestern Taiwan, Sci. Total Environ, 156460, https://doi.org/10.1016/j.scitotenv.2022.156460, 2022.

**Line specific comments**

*66-67 Change "practical" to partial*
Response: Thank you, we have revised the typo.

*81 Ejarque et al is not correctly cited in this sentence. The study was not a subtropical lake with typhoons. It was a mountain lake in the European Alps. However, the study is very relevant to this work and should be discussed in relation to the results in the discussion section.*
Response: Thank you for your suggestion. We have removed the citation and added some sentences about this paper in discussion.

*110-111 More information is needed to understand what was measured. What wavelengths were used to measure QSE? If this was an in-situ measurement, how were the results corrected for particle, temperature interreferences? What instrument was used?*
Response: Thank you for your comment. We have added the specific wavelength (254 nm) in this sentence.

*134-135 Was the portable fluorometer was used to measure Chl a? This should be specified.*
Response: Thank you, we have added the wavelength in the sentence.

*154-156 Water level was measured at a single river input. Was discharge also measured? How was water input estimated for the many other rivers? Was direct precipitation over the lake surface area also accounted for?*
Response: We used the storage function model to estimate the river discharge using precipitation over the inflow river basin, and Nash–Sutcliffe model efficiency coefficient for the water level was > 0.70; thank you.

*157-160 35% doesn't seem correct given the other precipitation values reported in the sentence. It also doesn't appear to match table 1 values.*

Response: The 35.6 % is via total typhoon rainfall (2,254 mm) over the total precipitation (6,332 mm) from 2015 to 2016. Sorry to make you confused

*Equations 1-6 Consider adding a table that clearly identifies each term, its units, and whether or not it was measured or fitted. The many terms are difficult to follow and are not immediately explained in the text before new equations are introduced.*
Response: Thank you for your suggestion. Table 2 was added to explain the terms and units of Equations 1-6.

*211 What is meant by absorption coefficient in units per day? Light attenuation typically has units per length.*
Response: Thanks for your comment. The alpha_PU and alpha_PL are constants to obtain the absorption rates via Chlorophyll *a* concentrations, which are not the light attenuation (Table 2-3).

*215-216 What type of regression? Linear/nonlinear?*
Response: It is a multiple linear regression. We have revised it; thank you.

*247 248 This sentence compares two periods of typhoon years.*
Response: Yes, we have revised the sentence; thank you.

*281 "perfectly" is subjective. Quantify this comparison including errors.*
Response: Thank you, we have removed "perfectly" from the text.

*286 Here alpha is referred to as a photosynthetic absorption rate, not a coefficient.*
Response: It is a coefficient, not the absorption rate. We have revised it; thank you.

*356-357 My understanding from the text above was that a mass balance was applied to remove the influx of riverine DOC from the NEP calculation. Otherwise, river inputs would dominate over autochthonous NEP in this small system. If riverine C is included in the NEP, then maybe NEP is not a good term for this model output. Perhaps it is better referred to as a DIC/DOC flux from mass balance.*
Response: Thanks for your suggestions. We have revised the terms thoroughly.

*710 I do not understand what is meant by "nonseasonal data" in this context. Can you use a different term?*
Response: Thank you for your comment; we have changed "nonseasonal data" to "inter-annual data" in the manuscript.

*Figure 6 should include confidence intervals for the daily modeled values.*

Response: "Best-fit" means the best result for model fitting, so the data would not have a confidence interval. Nakayama et al. (2022) also have the best-fit results in figure 7. Thus, we cannot add the confidence interval in Figure 6; sorry about that.

Reference:

Nakayama, K., Kawahara, Y., Kurimoto, Y., Tada, K., Lin, H.-C., Hung, M.-C., Hsueh, M.-L., and Tsai, J.-W.: Effects of oyster aquaculture on carbon capture and removal in a tropical mangrove lagoon in southwestern Taiwan, Sci. Total Environ, 156460, https://doi.org/10.1016/j.scitotenv.2022.156460, 2022.

---

## Author Response (AR2)

**Report #1:**

*The manuscript titled "Conceptual models of dissolved carbon fluxes considering interannual typhoon responses under extreme climates in a two-layer stratified lake" aims to discuss the impacts of extreme typhoon climates on the distribution of carbon (C) in small subtropical lakes, taking the Yuan-Yang Lake (YYL) as an example. The manuscript highlights how typhoons rapidly introduce significant amounts of terrestrial C into the lake, influencing the water chemistry in the lake. The study develops a conceptual dissolved C model and proposes that the loading of allochthonous C and river inflow intrusion affect the distribution of dissolved inorganic C (DIC) and dissolved organic C (DOC) in the lake under extreme climate conditions. This is an interesting study, but some modifications are required.*

Response: Thanks for your comments. The manuscript has been revised, taking into account your comments below.

**General comments:**

*More references should be included in the Introduction section on how extreme climate will impact the water chemistry in both small and large lakes. (e.g. 2023 Water Research, 10.1016/j.watres.2022.119448 and 2020 Water Research 10.1016/j.watres.2020.116471)*

*There are also some grammar and terminology issues to be addressed.*

Response: Thanks for your comment. We have revised these sentences by adding new references and have tried to clarify them as you suggested (lines 85 to 88).

**Specific comments:**

*Figure 1 I can't see any useful information on how river intrusion will change the upper and lower DOC-DIC from this figure. The allochthons C shown in the lower panel seem a little bit too short. Technically, the inflowing river mouth areas are significantly influenced by the rainstorm-induced C inputs. Also, only typhoons are discussed in the manuscript and I think the other extreme climate examples should be removed. The figure needs to be reorganized.*

Response: Thanks for your comments. We have removed the figure.

*Figure 2. Inflowing rivers should be included in this figure.*

Response: We have enlarged the marker as much as possible. The red arrows show the inflowing rivers. Thanks.

*Figure 3 I can see much higher Chl-a in the lower than in the upper layer of this lake.*

*Higher phytoplankton biomass is usually found in the upper layer of a specific lake. Please double-check your data.*

Response: Thanks for your comment. We have double-checked the data set to make sure it is correct. YYL is a humic and oligotrophic lake, resulting in the low Chl. a concentration in the upper layer (Tsai et al. 2008), but we sometimes found brief agal blooms in the lower layer around the end of March.

*Line 58: It should be written as fluxes.*

Response: We have revised it accordingly, thank you.

*Line111: It should be written as Fig. 2.*
*Line 149: It should be written as Fig. 1.*
*Line 160: It should be written as Fig. 2.*
*Line 284: It should be written as Fig. 6c.*
*Line 752: It should be written as Fig. 8.*

Response: We have revised these abbreviations, thanks.

*Line139:The specific wavelengths used were 430 nm (blue) and 662 nm (red). Was the portable fluorometer used to measure Chl-a? Please reorganize the text.*

Response: Thanks for your suggestion. We have rephased the sentence (lines 146 to 149).

*Line 246: Please change Lower NEPDOC in the formula, and pay attention to your language.*

Response: We have adopted the correct symbol, thank you.

**Report #2:**

*I think this is a nice contribution that couples measurements + modeling on an important issue of quantifying carbon cycling in lakes globally. However, before publication, I think there is still some lacking organization, synthesis of clear results, and connection to the study's broader objective. I have some suggestions and concerns that fall largely into the following categories that the author could consider incorporating:*

Response: I appreciate your positive and constructive comments. The manuscript has been revised, taking into account your comments below.

**1) Introduction edits:**

*Improve the flow to really synthesize the importance. There is also a harsh transition to talking about precipitation in line 80, I think you can weave concepts a little more strategically. From my interpretation, I see 4-5 paragraphs in the introduction addressing the following: we need to keep improving quantifying carbon flux in lakes globally, it is not clear how climate will impact these fluxes, extremes in precipitation are likely to increase in a lot of parts of the world (likely in this study site as well), and it's been shown that precipitation has impacted carbon cycling in lake systems, but it remains foggy. So, in this study….*

Response: We appreciate your comments. We have added one more paragraph to introduce how typhoons impacted DIC and DOC in subtropical shallow lakes (lines 82 to 94).

**2) Results edits:**

*I think the results section still remains hard to follow, it was suggested from a previous review that the descriptor "nonseasonal" is confusing. I agree and I think it still remains in the manuscript and hasn't been changed all over (maybe just a few places). If you are also going to use seasons, should you consider using a hemisphere descriptor for clarity (for example, boreal fall for Northern Hemisphere fall…)? I would maybe suggest being clearer about your section headers. Maybe divide into typhoon and non typhoon years and then into measured and modeled results. I think right now it goes back and forth a lot and is hard to follow. I think it will also improve when the term "nonseasonal" is removed.*

Response: Thanks for your comments. "Nonseasonal" was confusing, as you pointed out. We have therefore thoroughly removed the results for nonseasonal data.

**3) Figure edits:**

*Fig 1: In conceptual model, can droughts not also impact lake thermal stratification? Why is DIC connected to photosynthesis and respiration but not DOC? I believe the DOC pool is influenced by those processes? How does the bar of different C pools (auto/allocthon) on the bottom fit into the conceptual model? For example, sediments are in blue like autochthonous C, but it is likely your sediments are a mixture (likely dominated) by allochthonous C too. There are also additional biogeochemical processes + C flux from sediments that aren't addressed in the figure (but are brought up in the conclusion)...*

Response: Thanks for your comments. We agree that the figure was confusing. We have removed it.

*Fig 2: I would make your markers bigger on the map.*
Response: We have enlarged the marker as much as possible. The red arrows show the inflowing rivers. Thanks.

*Fig 3: should you maybe just write the p values on the respective graph panels instead of the key to the values (ns, \*, \*\*)? For example, since you use the term "nonseasonal" I think ns means that, but I do not believe that is the case...*
Response: We have added the *p*-values in the graph panels. Thank you.

*700-702: I see no red values in this table.*
Response: We have removed it from the manuscript. Thank you.

*706: Perhaps rewrite figure caption to: "Conceptual diagram of river intrusion (red arrows) and thermal stratification 706 (red dashed line) influencing dominant responses of DIC and DOC in a subtropical two-layer 707 stratified lake under extreme climates" Also the tables in the format (as is) are not well aligned (e.g, titles on two lines, hard to read), you might need to adjust. Maybe add your "Level" characterizations to Table 2 or 3 - the differentiation is getting a bit lost in the methods.*
Response: We have removed the figure and added the Level characterizations to Table 2.

**4) Discussion: consider**
*Sub-organizing the discussion into typhoon/non typhoon years, or model/measured data, or how the modeled/measured data disagree? I think this will help add flow that feels missing as is. Reviewer #1 suggested adding some additional sentences on*

*seasonal CO2 emission flux, but it is not popping out to me right now...*

Response: Thanks for your suggestions. We have added the subtitle to sub-organize the discussion and have added some sentences to discuss the $CO_2$ emission in this study (lines 383 to 394).

***5) General specifics***

*to be added: I mentioned a few line specific places where I think you can just provide a bit more detail to give the reader clarity, but there are other places throughout the manuscript that can benefit from this. A sentence that I marked but can be used as an example: "The water depth is not only steady but also changes"< I think additional detail is needed to clarify this. Another in the introduction: For example: "River inflows, sediment, and respiration contribute to DIC loading into lakes">> sediment what? Flux? Respiration in the water column or post depositional biogeochemical processes? There are a myriad of steps that exist between...*

Response: Thanks for your comments. We have revised these sentences and tried to clarify them as you suggested.

***Line specific comments:***

*Line 21: models to model*

Response: We have made this change. Thank you.

*Line 24: should transportation be transformation?*

Response: Yes, we have replaced the word. Thank you.

*Line 28: change a load to loading*

Response: We have corrected the grammatical error. Thanks.

*Line 39: fresh or salty water columns- can you note?*

Response: Thanks for your comments. We focused on freshwater ecosystems, so this refers to freshwater columns.

*50: with to within*

Response: We have corrected the error. Thank you.

*Line 80: big scale shift to talking about rain– expand on what is known about rainfall impacting DOC and C cycling...*

Response: Thanks for your suggestion. We have revised this paragraph.

*121-123: what nutrients? What organisms– can you be more specific?*

Response: Thanks for your suggestion. We have revised the wording to make it more specific.

*183/184: Fix sentence "The water depth is not only steady but also changes"> also is this different font?*

Response: We have removed the sentence and revised the font. Thank you.

*233: different font*
*255: different font*

Response: We have revised the subtitle fonts. Thank you.

*256-258: clarify this sentence– what is the difference between typhoon and non typhoon years here, and what is this result trying to compare or conclude?*

Response: We have revised the sentence, thank you.

*261: be more specific- what are you comparing in this t test?*

Response: We have revised the sentence, thank you.

*275-276: should this conclusion (correlation with resp, DIC and chl) be in the discussion? Or give a reference for this interpretation…*

Response: Thanks for your suggestion. Because we had investigated this before, we added references (Lin et al. 2021 and Tsai et al. 2008) to this sentence.

*282: define NSE*

Response: As in equation (8). Thank you.

*285: best fit different text*

Response: We have revised this, thanks.

*298-299: is this a statistical output or an interpretation?*

Response: It is an interpretation of parameter values. We have revised it. Thank you.

*311-312: Can you clarify "the upper layer DIC and DOC consumed approximately 3.7 times more DIC in the typhoon years than in the non-typhoon years"? How does DIC and DOC consume, maybe missing biogeochemical steps here….*

Response: Thank you for your suggestion. We have added some sentences about that in Sect. 4.1 (lines 380 to 394).

*323: in saying "upper layer ratios" are you referring to DIC:DOC in the upper layer? Likely should clarify....*

*326: You talk about ratios varying within this paragraph, but only use a value at the very end in regards to the lowest... keep that consistent throughout the paragraph if you'd like to compare values/numbers.*

Response: We have removed this paragraph because it did not show necessary information for this study and because the sentences were confusing, as you pointed out.

*341-345: more specifically describe the result or point of this result.*

Response: We have added more sentences to describe Figs. 8-9 (lines 342 to 351). Thank you.

*351: be more specific about what parameters changed. What do you consider water quality data? How might remineralization change with anoxic conditions.*

Response: Thanks for your comment. We have revised the wording and described the change with anoxic conditions (lines 359 to 367).

*355: probably good to specifically reference microorganisms rather than just organisms here?*

Response: Thanks for your suggestion. Indeed, the use of "microorganism" in this sentence (line 362) is well advised.

*365: corresponds or proves hypothesis?*

Response: "Corresponds" is correct. Thank you.

*372-375: Can you clarify the following: "Additionally, 372 because of the absence of typhoon-induced mixing and allochthonous C loading, the absolute 373 values of total fluxes in the non-typhoon years were less than those the non-typhoon years 374 (Table 4). "*

Response: Thanks for your suggestion. We have added $CO_2$ emission and other flux data to support this sentence (lines 383 to 394).

*377: seasonal C what? Concentration? Flux? Sequestration?*

Response: Concentrations. We have clarified this. Thank you.

*408: Remove the phrasing "to our knowledge"*

Response: We have removed the phrase, thank you.

*414: two commas*

Response: Thank you, we have fixed the problem.

*420: Typhoon disturbances or seasons?*

Response: Typhoon disturbances. We have clarified the sentence. Thank you.

*421: Rephrase sentence:" Without typhoons, the strength of thermal stratification was the primary determinants (determinant of) the seasonal and interannual patterns of DIC and DOC concentration. Typhoon-induced upwelling and loading facilitated 102.2 mg-DIC m-3 d-1 and 62.3 mg-DOC m-3 d-1 flux in YYL, respectively."*

Response: We have rephrased the sentences. Thanks for your suggestion (lines 435 to 438).

---

## Author Response (AR3)

**Report #1:**

*Fig. 1 Yuang-Yang Lake in the lower panel and Yuanyang Lake in the right panel should be changed to Yuan-Yang Lake.*

Thanks for your comment. We have revised the text in Fig. 1.

**Report #2:**

**General comments**

*I appreciate the changes and improvements that were made to this manuscript, but I still think additional editing needs to be completed before acceptance for publication. In this second round of peer review, I believe that the results section and amount of figures is too long, and should be compressed to detail the most meaningful results. As written, I find the results still extremely confusing to get through, and find it hard to streamline what is important and how the authors reach the conclusions from the figures and results presented as is.*

We appreciate your constructive comments. The manuscript has been revised, taking into account your comments below.

*I would suggest additional time in re-structuring the results. Firstly, I still don't understand how you talk about the measurements (observations) and the simulated (conceptual model) data. It seems like most of your data is simulated, but you still made measurements- can you differentiate more clearly?*

Thanks for your comment; we have restructured the results section thoroughly and attempted to depart the results of measurements (section 3.1) and models (sections 3.2 and 3.3).

*Right now, I have also found that many of the statements seem redundant and this flows into the discussion. Much of the discussion just reads like a continuation of results, not an actual evaluation of the results in the context of how it's expanded what is known about the current literature/biogeochemical processes in subtropical lakes. I would suggest spending more time discussing specific results as they tie to specific or broader patterns in biogeochemical processes, and removing some of the truly analytical results (perhaps moving those into the results section, like lines 384-391). So, my overall analysis would be to compress the results and figures and expand the discussion as it relates to biogeochemical processes.*

Thanks for your comment and suggestion. We have moved these sentences to the results section and added and revised some paragraphs to discuss the biogeochemical processes as your suggestion (Lines 377–390 and 433–441).

*I also think you can move some of your figures into supplemental. In my opinion, there is a lot of redundancy (like figures 5 and 6 without clearer explanation of Level 1/2) and figure 4 doesn't seem to add much as a priority figure. Can you consider more succinctly combining some of the findings into figures that make things easier to follow? For example- do you think Figure 7 and 8 are both necessary? Figure 9 comprehensively brings together observations and modeled values (I like this). Is Figure 3 populated by observations or modeled values?*

Thanks for your suggestion. We have combined Fig .5 and Fig. 6 to new Fig. 5 and moved Fig. 7 and Fig. 8 to supplement (Fig S1 and Fig S2).

*Other conceptual things that need clarity:*

*What do you mean by interannual variability: are you just referring to seasonal change? For example, how is 3.4 different than 3.1? Can you clarify within section 3.3 too?*

Thank you for your comment. Interannual variability means the seasonal dynamics of the two years' data. Also, we have restructured the results section.

*Can you clarify/write in more detail your methods of sample collection? Did you collect separate samples for DOC that was not filtered on the GFF filters? What did you use the 0.7 um particulates for? That is too big of a mesh size for DOC concentrations.*

Indeed, the standard technique for analyzing DOC in freshwater samples is based on a filter paper with a pore size of 0.45 μm. However, some DOM research community pointed to the pore sizes of 0.45 and 0.2 μm sometimes are constructed from organic substances such as cellulose acetate, that might potentially induce contamination (Denis *et al.*, 2017). Thus, the 0.7 μm glass fiber filters might be better to measure the DOC concentration.

Denis, M., Jeanneau, L., Pierson-Wickman, A. C., Humbert, G., Petitjean, P., Jaffrézic, A., & Gruau, G. (2017). A comparative study on the pore-size and filter type effect on the molecular composition of soil and stream dissolved organic matter. Organic geochemistry, 110, 36-44. https://doi.org/10.1016/j.orggeochem.2017.05.002

*Did you only do some of the samples for water quality once? And then the outflow monthly? Maybe include the number of observations on your figure/within your figure caption.*

All the water samples were collected monthly, but sometimes, we could not trip to YYL because of the breeding season in April or other accidents, such as tree snaps and landslides. We have added sampling numbers in Fig. 1 and Fig. 5 caption as your suggestion, thank you.

*Can you please provide further clarity in the text on what is Level 1 and 2 and what climate extremes does it relate to? Localized extremes? Already known modeled extremes?*

Thanks for your comments. We followed buoyancy frequency (Brunt–Väisälä frequency) in YYL (Lin et al. 2021), and expected extreme weather events might induce stronger seasonal thermal stratification from spring to summer and longer overturns from autumn to winter to determine the climate extreme scenarios.

*Another read through to clarify (concepts + references) and condense sentences. An example where sentence to me was hard to follow:*

*369: "Therefore, these physical and biogeochemical processes might describe different patterns between the upper and lower layers (Fig. 4). "In summer, the spatial differences*

*between layers in DIC and DOC were inhibited due to strong thermal stratification, describing the positive upper net primary production and lower negative net primary production (Lin et al., 2021)."*

*What specific patterns? What would we expect (provide reference)? Spatial differences between layers in DIC and DOC< what does that mean? Across a transect? Within layers? How is it describing the positive/negative primary production. Missing a link between DIC/DOC variability and PP....*

In the paragraph, we have added more sentences to describe biogeochemical and physical processes clearly. That can help us gain deep insight into this study. Thank you. (Line 377–390 and 433–441)

**Specific comments**

43: Reference at the end of that sentence?

45: important for humans because of the C processing? Or because of C processing providing availability to foodwebs that support human resources? Alternatively, you could mean C storage? < Consider making a better connection there.

Thanks for your suggestions. We rephrased the sentences. (Line 45–47)

49: Start a new paragraph when you begin talking about small lakes.

We revised the paragraph by separating paragraphs as you suggested; thank you for your comment. (Line 48)

56: Remove however

We removed the word, thank you.

66: Consider rephrase "Not only taking is taking in situ measurements difficult, but resolving the dynamics and interactions…. Remains complex."

Thank you for your comment; we have removed the sentence.

102: was that a different year than 2004, can you clarify?

We added the period (in the summer and autumn of 2015) in the manuscript; thank you. (Line 100)

122: remove second its

We have removed it; thank you.

128: the world annually (half of totally precip in YYL annually)

Thank you for your suggestion. We have added world annually in the sentence. (Line 127))

140-144: please clarify these sentences. You collected water via van Dorn for various parameters (DOC/DIC/Chl.a). You also collected GFF filters for POC at the outflow which would be the filtrate– as 0.7 represents POC not DOC– can you clarify? So, you had liquid samples and filtrate, correct?

Thanks for your comments. We filtrated water samples and used filter paper samples to obtain DIC/DOC and Chl. a concentrations. However, some DOM research community pointed to the pore sizes of 0.45 and 0.2 μm sometimes are constructed from organic substances such as cellulose acetate, that might potentially induce contamination (Denis *et al.*, 2017). Thus, we used 0.7 μm glass fiber filters might be better to measure the DOC concentration.

Denis, M., Jeanneau, L., Pierson-Wickman, A. C., Humbert, G., Petitjean, P., Jaffrézic, A., & Gruau, G. (2017). A comparative study on the pore-size and filter type effect on the molecular composition of soil and stream dissolved organic matter. Organic geochemistry, 110, 36-44. https://doi.org/10.1016/j.orggeochem.2017.05.002

148-153: Provide more detail/clarity here in what you did. What did the fluorometer measurements give you? What machine was used to measure Chla after methanol extraction?

Thanks for your comments. We used a portable fluorometer (model 10-AU-005-CE; Turner Designs, Sunnyvale, CA, USA) to estimate Chl. a concentration. We have revised the sentences. (Line 148–152)

152-153: do you mean all analysis was complete within 72 hours of exposure to light to reduce the degradation?

We have added the sentence in the paragraph; thanks for your suggestion. (Line 151–152)

159: Add a sentence in how the observable data was used within the conceptual equations model– how did you make that link?

We added a sentence to link the paragraphs; thanks for your suggestion. (Line 158–160)

161: extra dash there.

We removed it; thank you.

165: remove one of the uses of meteorological

We removed the text.

172: did you sample the secchi disk depth at a certain interval in that time frame?

We added more information in the manuscript; thanks for your comment. (Line 171–172)

173: How did you confirm there were four strong typhoons recorded? Wind speed/other meteorological parameters? Clarify.

Thanks for your comment. Four strong typhoons were recorded by using wind speed and rainfall meteorological parameters (Table 1). We added more explanations in the manuscript. (Line 177–178)

173-174: Please clarify this concept. It is not 35.6% of annual rainfall, it's 35.6% of rainfall across 2 years of typical rainfall (>3000 mm/y).

We revised the sentence to rainfall across two years of typical rainfall. Thank you. (Line 173–175)

176: So, 2017 and 2018 are both below average years? ~1268 mm/y instead of 3000 mm/y as suggested in line 127?

Thanks for your comment. The rainfall was recorded around 1800 to 4500 mm yr$^{-1}$ from 1995 to 2005 in YYL, depending on annual typhoon numbers (Lai et al. 2006). Yes, the annual rainfall from 2017 to 2018 was below average. We revised the sentence in the manuscript. (Line 176–178)

177: Was average water depth higher in 2015/2016?

We added more information in the manuscript; thanks for your suggestion. (Line 176–179)

178-179: In general, please clarify how you defined typhoon/non typhoon years at the beginning of this paragraph. The way it is written makes me think the rainfall in 2015 and 2016 is less than 2017/2018 (which you are considering non typhoon), I think you need to be really clear that it's not rainfall alone, but it is temporal variability of the rainfall (falls all between X months), avg wind speed higher, discharge overall lower, etc.

We added the annual averages of wind speed in the paragraph and Table 1; thank you. (Line 173–180)

194: variated to varied

We revised the text; thank you. (Line 196)

197: add "was used to establish"

We added it to the manuscript; thank you. (Line 199)

205: missing parentheses

We added the parentheses in the manuscript; thank you. (Line 207)

219: Start new paragraph when talking about climate change.

Thanks for your suggestion. We separated the paragraph as you suggested. (Line 222)

224: what were the extreme conditions based on? Qin? how did you define Level 1 and Level 2 respectively? How can we compare this to real-time projections of what is expected in climate scenarios for the YYL– rainfall amounts?

Thank you for your comment. We expected that "Extreme weather events might induce stronger seasonal thermal stratification from spring to summer and longer overturns from autumn to winter, thereby changing C distribution and transportation within water bodies (Kraemer et al., 2021; Olsson et al., 2022a; Woolway et al., 2020)". Thus, we changed the QU and QL by using the ratios of Qin (Table 2) to test our river intrusion hypothesis. Thus, we used the river discharge and considered buoyancy frequency (Brunt–Väisälä frequency) to obtain the projections.

267: I am confused- are these results modeled or measured?

Thanks for your suggestion. We revised the subtitles in the results. The measurement results were shown in section 3.1, and the simulation results were shown in sections 3.2 and 3.2.

299-300: can you clarify what you are comparing here- DIC measured/modeled? Best fit to what sort of regression/relationship?

Thank you for your comment. We used NSE (Nash–Sutcliffe model efficiency coefficient) and $R^2$ to know the robustness of how the model fits the field observations to find the best-fit conditions. We revised the sentence.

310-311: Can you clarify how you got to that conclusion? What did you compare- state it directly?

We removed the sentence as the sentence does not show the conclusion clearly; thank you.

318: Was this relationship between Fc and Flux DIC in both typhoon and non typhoon years– consider rephrasing the sentence as you are talking about both?

We rephrased the sentence; thanks for your suggestion. (Line 330–331)

320-321: consider rephrasing sentence: two 'declines' in a short sentence.

We replaced one of the 'declines' in the sentence; thank you. (Line 333–336)

322: What's with the quotations? The text size also looks different…

We removed the quotations and corrected the text size.

You start section 3.4 with mentioning these are simulated results, but seemingly the references to figures in the above sections are also the conceptual model simulations. I find this confusing. Streamline what is observed and what is modeled.

Thanks for your comment. We have restructured the results and attempted to separate the results of measurements (section 3.1) and models (sections 3.2 and 3.3).

328/329: looks like different text size

We corrected the text size; thank you.

359: As Table 1 shows, four strong typhoons were recorded, contributing a total of 2,254 mm 173 of precipitation in all 24 months of 2015 and 2016, This accounted for 35.6% of the annual 174 precipitation. However, no typhoon rainfall was recorded at YYL in 2017 and 2018; the total 175 precipitation in that 2-year period was around 2,537 mm. << how does this compare to what is said around lines 173?

We revised and added the correct data (35.6%) in the manuscript; thank you. (Line 365)

410-413 repeating the concept in 367-369

We removed the sentence; thank you.

443-445 I would rephrase this. I don't think the words autochthonous or photo biochemical is used efficiently here. (e.g., within lake, primary production, photochemical degradation might be better choices to incorporate)

Thank you for your suggestion. We revised the sentence. (Line 458–460)

---

## Author Response (AR4)

**Editor comments**

*Thank you for your diligent work in addressing three rounds of significant revisions. The revised manuscript has shown significant improvement. I am pleased to accept it for publication. There are a few minor technical corrections that require attention, see below:*

Dear Dr. Shen (Associate Editor),
Many thanks for your diligent handling of this, we received many constructive and inspiring comments and suggestions for this paper from reviewers and much help us. The manuscript has been revised, taking into account your comments below.

*line 38: combine double sets of parentheses*
We have revised this, thank you.

*line 266 "Measuring data" implies an ongoing process of data collection. I think the author meant "Measurement data" or "Measured data".*
Thanks for your correcting, we have revised this grammatical error.

*line 485 Data availability: Please be more specific about which data were taken from these references. We recommend that any data set used in your manuscript is submitted to a reliable data repository and linked from your manuscript through a DOI.*
Thanks for your suggestion, we have added the DOI link for database in the data availability section.